# Progress toward Polymerization Reaction Monitoring with Different Dienes: How Small Amounts of Dienes Affect *ansa*-Zirconocenes/Borate/Triisobutylaluminium Catalyst Systems

**DOI:** 10.3390/polym14163239

**Published:** 2022-08-09

**Authors:** Amjad Ali, Jamile Mohammadi Moradian, Ahmad Naveed, Tariq Aziz, Nadeem Muhammad, Chanez Maouche, Yintian Guo, Waleed Yaseen, Maria Yassen, Fazal Haq, Mobashar Hassan, Zheqing Fan, Li Guo

**Affiliations:** 1Research School of Polymeric Materials, School of Materials Science & Engineering, Jiangsu University, Zhenjiang 212013, China; 2MOE Key Laboratory of Macromolecular Synthesis and Functionalization, Department of Polymer Science and Engineering, Zhejiang University, Hangzhou 310027, China; 3Biofuels Institute, School of Environment and Safety Engineering, Jiangsu University, Zhenjiang 212013, China; 4School of Engineering Yunqi Campus, Westlake University, Hangzhou 310024, China; 5Department of Environmental Engineering, Wuhang University of Technology, Wuhan 430223, China; 6Zhejiang Hetian Chemical Co., Ltd., Hangzhou 310023, China; 7School of Chemistry and Chemical Engineering, Jiangsu University, Zhenjiang 212013, China; 8Department of Chemistry, Gomal University, Khyber Pakhtunkhwa 29220, Pakistan

**Keywords:** metallocene, polymers active centers, ethylene, propylene, 1-hexene, propagation rate constant

## Abstract

The objectives of this work were to address the fundamental characteristics of ansa-zirconocene catalyzed E/diene copolymerization and E/diene/1-hexene and E/diene/propylene terpolymerizations, and the quantitative relationship between diene structure and polymer chain propagation rate constant in term of quantifiable catalytic active sites. One of the most important but unknown factors in olefins ansa-zirconocene complexes is the distribution of the catalyst between sites actively participating in polymer chain formation and dormant sites. A set of ethylene/dienes copolymerizations, and ethylene/dienes/1-hexene and ethylene/dienes/1-hexene terpolymerizations catalyzed with *ansa*-zirconocenes/borate/triisobutylaluminium (*rac*-Et(Ind)_2_ZrCl_2_/[Ph_3_C][B(C_6_F_5_)_4_]/triisobutylaluminium (TIBA) were performed in toluene at 50 °C To determine the active center [C*]/[Zr] ratio variation in the copolymerization of E with different dienes and their terpolymerization with 1-hexene and propylene, each polymer propagation chain ends were quenched with 2-thiophenecarbonyl, which selectively quenches the metal–polymer bonds through acyl chloride. The ethylene, propylene, 1-hexene, and diene composition-based propagation rate constants (*k_p_*E, *k_p_*P, *k_p_*1-H, and *k_p_*diene), thermal (melting and crystalline) properties, composition (mol% of ethylene, propylene, 1-hexene, and diene), molecular weight, and polydispersity were also studied in this work. Systematic comparisons of the proportion of catalytically [Zr]/[C*] active sites and polymerization rate constant (*k_p_*) for *ansa*-zirconocenes catalyzed E/diene, E/diene/1-hexene, and E/diene/propylene polymerization have not been reported before. We evaluated the addition of 1-hexene and propylene as termonomers in the copolymerization with E/diene. To make a comparison for each diene under identical conditions, we started the polymerization by introducing an 80/20 mole ratio of E/P and 0.12 mol/L of 1-hexene in the system. The catalyst behavior against different dienes, 1-hexene, and propylene is very interesting, including changes in thermal properties, cyclization of 1-hexene, and decreased incorporation of isoprene and butadiene, changes in the diffusion barriers in the system, and its effect on *k_p_*.

## 1. Introduction

Catalysis is purely a kinetic phenomenon; kinetics regulate the molecular weight, polydispersity, comonomers insertion, microstructure, and bulk properties of polymers synthesized by group 4 transition metal catalysts [1,2,3,4,5,6]. However, polymers properties depend on several factors: the nature of catalyst speciation, type, and nature of monomers, reaction temperature, molecular weight, microstructure, and polydispersity index (Đ), comonomer content, stereotacticity, amount of cocatalyst, and loaded catalyst [7,8,9]. In this regard, the properties of polyolefin and dipolyolefins with heterogeneous Ziegler–Natta catalysts demonstrate precise control of polymer topology, thermal behavior, and composition, but produce broader molecular weight distribution and multiple types of active sites [10,11,12,13]. The advent of homogeneous single-site metallocene catalysts for olefin and diolefin polymerization has revolutionized the industrial production of polyolefins and diolefins [14,15,16,17]. In addition, the living coordination polymerization of olefins, in which chain propagation (growth) continues in the absence of chain termination, offers them opportunities to explore and design new fundamental types of polyolefin. However, given the previous impact of single-site metallocene catalysts on polyolefin research and manufacturing, it is surprising that the basic element of catalytic olefin reactions have been firmly established over the past three decades. There are still no well-defined kinetic phenomena for the fundamental routes for quantifying catalyst speciation and kinetic data, such as chain initiation, propagation, transfer, and chain termination, of a single-site metallocene-catalyzed olefin and diolefin polymerization reaction with the desired activity [18,19,20,21,22]. In addition, the lack of well-defined rate laws reflects difficulties posed by higher catalytic activities and the extreme sensitivity of metallocene catalysts to impurities. More preventive is the problem of determining the fraction of catalyst species actively involved in producing polymer chains instantaneously, such as active site count. Although efficient methods require improvements in understanding active site counts and accelerating innovation in metallocene-catalyzed olefin polymerization technology. Without knowledge of the speciation and concentration of catalytically dynamic active species, it is impossible to correlate molecular mechanisms with rate laws [18,23,24].

The production of polyolefins exceeds 100 million tons annually; there is no exception. Some detailed kinetic studies proposed that 100% of the metallocene catalyst added to polymerization is active during the reaction [25,26]. Only a few catalysts exhibit 100% active sites; however, in some cases, the kinetic analysis found less than 100%, even below 5% [27,28]. Clark R. Landis et al. have explored extensive variations in the active site concentration amongst different group 4 metal-based catalysts used in olefin polymerization [29,30]. Guo et al., using the same metallocene catalyst with varying active site counts, recommended that variation in active site count depends on various reaction parameters, such as monomers ratio, activators, additive temperature, and polymerization time [31]. Several direct and indirect procedures to estimate the total number of active site count or catalyst–polymeryls, have been investigated, largely involving quench-labeling methods. However, the direct observation of active site count requires spectroscopic techniques and provides the most fascinating information for mechanistic studies of polyolefin; still, collection of the data is hampered by the requirements for specialized apparatus and high catalyst concentrations [8,32,33,34]. Furthermore, an indirect procedure for counting active sites count has been established through calculation, depending on gel chromatography data (Mw data), labeling investigation, and killing experiments [3,35]. In contrast, Moscato et al. [28] developed a more convenient quench-labeling agent method to determine the active site count for sensitive metallocene-catalyzed olefin polymerization. Our group sought to augment the quench-labeling agent method of Moscato and D. Luke Nelsen et al. [35] by using 2-thiophenecarbonyl chloride (TPCC) quench-labeling agent, which is more is a convenient reagent for identifying the active sites counts in the E and propylene homo- and copolymerization with a-olefins (propylene, 1-hexane) and olefins (ENB, VNB, and VCH) catalyzed by (*rac*-Et(Ind)_2_ZrCl_2_/TIBA/borate and (*rac*-Et(Ind)_2_ZrCl_2_/MMAO [15,36,37]. TPCC selectively quenched the metal–polymer bonds through acyl chloride, which has been verified in the polymerization of olefins with Ziegler–Natta, nickel–diamine catalysts and metallocene/MMAO.

In this work, we seek to strengthen our understanding of critical issues in the polymerization of E with a-olefins and diolefins through fundamental kinetic investigation. We demonstrate the application of 2-thiophenecarbonyl chloride quench-labeling to count the active sites counts to determine the propagation rate constant (*k_pP_E*, *k_p_E*, *k_p_P*, *k_p_1H*, *k_p_Dienes)* for the E/diene, E/diene/1-hexene, and E/diene/propylene co- and terpolymerization catalyzed by *rac*-Et(Ind)_2_ZrCl_2_/[Ph_3_C][B(C_6_F_5_)_4_]/TIBA. This catalyst system is used in E/P/diene terpolymerization and the industrial production of polyethylene. Because of its prominence as a commercial polymerization catalyst, M-II/TIBA/borate-II has been the subject of extensive study. With the effect of liners IP, BD, HD, and an acyclic diene (ENB, VNB, VCH) along with propylene and 1-hexene, on the active site counts, TPCC is shown to act as an efficient quencher to the reactive symmetrical metallocene. Further examination of co- and termonomer content, molecular weight, polydispersity index, chain transfer, termination, and chain propagation rate constant demonstrates the mechanistic complexity and existence of significant catalyst dormancy with different types of dienes under identical reaction conditions.

## 2. Materials

The polymerization catalyst *ra*c-Et(Ind)_2_ZrCl_2_ (Sigma-Aldrich, St. Louis, MO, USA) and triisobutylaluminum (TIBA) were bought from Albemarle Co. (Charlotte, NC, USA) and diluted in n-heptane according to previous study [15], while borate was donated by Sinochem Lantain, Hangzhou, China. Ethylene and propylene (99.9% purity) were purchased from Zhejiang Gas Mixing Co. (Hangzhou, China). Isoprene (IP), 5-ethylidene-2-norbornene (ENB), vinylcyclohexene (VCH), and VNB were from Aldrich, while butadiene was purchased from Zhejiang Gas Mixing Co., (Zhejiang, China) and dehydrated

## 3. Polymerization

All copolymerization and terpolymerization were performed in the 100-Ml Schlenk round-bottomed glass reactor under a nitrogen atmosphere. Before starting the polymerization, the glass reactor was purged with nitrogen and 50 mL of toluene (polymerization grade) was added. The metallocene complex and borate were prepared in 20 mL of toluene under a nitrogen environment and used for the set of experiments. For E/diene copolymerization, a pure form of E was added to the reactor, followed by additions of dienes and TIBA and run for 5 min, and then metallocene and borate activator to initiate polymerizations. However, for E/diene/1-hexene polymerization, ethylene with 0.12 mol/L of 1-hexene was added. In addition, the E/diene/propylene terpolymerization, E/P 80/20 molar ratio was added and followed the same method, addition of the diene, TIBA, metallocene, and borate to start the polymerization. After a planned time, the E/diene copolymers and E/diene/1-hexene and E/diene/propylene polymerization reactions were quenched with TPPC, which selectively quenched the metal–polymer bonds through acyl chloride, which has been verified by application in the E and P homo- and co-polymerization catalyzed with metallocene/MMAO, Ziegler–Natta/TIBA, and nickel-diimine catalyst systems [11,38]. In addition, further volume of ethanol with 2% hydrochloric acid was added to decompose the borate and metallocene catalyst. The obtained polymer product was further precipitated by adding 200 mL of ethanol. To completely remove the unreactive TPPC and other impurities from polymers, samples were thoroughly purified by a three-step process by dissolving 30 mg of polymer in 50 mL of octane at 120 °C, which was then precipitated in 200 mL of ethanol [11,31,39]. All polymer samples were dried using the vacuum drying system at 40 °C.

## 4. Characterization

The microstructure and composition of the obtained polymers were measured using a Varian Mercury plus 300 spectrometers (Varian Corporation, Palo Alto, CA, USA) (75 MHz) according to the literature [9,31]. Molecular weight and polydispersity of E/diene, E/diene/1-hexene, and E/diene/propylene were measured with high-temperature GPC using PL-gel MIXED-B columns and 1,2,4-trichlorobenzene as the eluent at a flow rate of 1.0 mL/min [40]. The thermal behavior of E/diene, E/diene/1-hexene, and E/diene/propylene polymerization was examined with a TA Q200 instrument calibrated with indium and water. First, the copolymer samples were heated to 150 °C, while terpolymers were heated at 130 °C for 5 min, and then cooled to 20 °C at a rate of 10 °C/min. After extensive purification of obtained polymers, an YHTS-2000 ultraviolet fluorescence sulfur analyzer was used to calculate the polymers’ sulfur (S) content according to our previous study methods [9,39]. However, a blank E/diene copolymer and E/diene/1-hexene and E/diene/P were prepared under comparable reaction conditions without the TPCC quenching step, which led to a sulfur concentration of 0 ppm.

## 5. Results and Discussion

Systematic comparisons of the proportion of catalytically [Zr]/[C*] active sites and polymerization rate constant (*k_p_*) for *ansa*-zirconocenes catalyzed E/diene, E/diene/1-hexene, and E/diene/propylene polymerization have not been reported before. We evaluated the addition of 1-hexene and propylene as termonomers in the copolymerization of E/diene. To make a comparison for each diene under identical conditions, we started the polymerization by introducing an 80/20 molar ratio of E/P and 0.12 mol/L of 1-hexene in the toluene solution at a temperature of 50 °C.

The sulfur content (S) of the purified E/diene, E/diene/1-hexene, and E/diene/propylene polymers samples were analyzed and calculated by analyzing three samples of each polymer to determine the catalytic [Zr]/[C*] active sites according to the following equation [S] = [C*] and finally taking the average of these tests. The ethylene, propylene, 1-hexene, and dienes composition-based propagation rate constants (*k*_p_) were also calculated according to our previous study [9,29,36,37,38]. The results of all co- and ter-polymerization including composition, catalytic [Zr]/[C*] active sites, thermal properties, activity, and monomers, are summarized in Table 1. The plots of activity with the addition of 1-hexene and propylene and their effect on ethylene incorporation rates are shown in Figure 1.

The catalyst behavior against different dienes, 1-hexene, and propylene, is very interesting, including changes in thermal properties, cyclization of 1-hexene, and decreased incorporation of isoprene and butadiene. It was found that the addition of butadiene and isoprene with ethylene slightly reduced the catalytic activity 2.26 × 10^6^ and 2.78 × 10^6^, but decreases in activities are more significant with VNB and VCH, considering that active catalytic sites are more functional for ethylene in the presence of butadiene and isoprene compared to VNB and VCH due to the less bulky structure and steric hindrance. In addition, the VNB and VCH more significantly prevent attachment of the ethylene monomer to the active catalytic sites than isoprene and butadiene. To overcome this issue, an appropriate amount of 1-hexene and propylene were added to the system to improve the polymerization activity. Moreover, in the polymerization of E/diene, with the addition of 1-hexene (0.012 mol/L) and the 20-mole ratio of propylene with ethylene, the polymerization activities were significantly increased due to the comonomer effect. Comparable results showing these effects were reported by Ahmadjo et al. [41], in the E/1-hexene and E/P polymerization catalyzed with heterogeneous Ziegler–Natta, metallocene/MMAO, and metallocene/MAO catalytic systems [15,31,41,42].

He proposed that the insertion rate of ethylene could not describe reduced or increased activities. Still, it is possibly described by the activation of catalyst doormat sites by dienes or 1-hexene, which will be explain in the next part of this paper.

In Figure 1 and Figure 2, the incorporation rate of ethylene decreased when VNB was added as the comonomer instead of VCH, but the presence of isoprene and butadiene resulted in a stable incorporation rate. In contrast, in the polymerization of E/diene/1-hexene, the incorporation rate of ethylene and dienes is reduced with 1-hexene, but the incorporation of 1-hexene is higher than dienes and lower than ethylene. Generally, dienes in co- or terpolymers are lower than ethylene. In E/diene/P terpolymers, the incorporation rate of dienes was increased compared to E/diene/1-hexene. In contrast, the incorporation rate of ethylene decreased. This catalyst system’s active sites are more active in inserting the dienes in the terpolymers in the presence of propylene. The effect of 1-hexene and propylene on the incorporation rates of ethylene is illustrated in Figure 2. During the E/diene copolymerization, a large amount of dienes was inserted, especially VNB 7.22 mole%, nearly equal to the incorporation of 1-hexene. In contrast, the effect of 1-hexene is more serious for isoprene and butadiene, and the incorporation rate is almost zero. 1-Hexene more significantly reduces the incorporation rate of isoprene and butadiene compere to VNB and VCH. In the case of E/diene polymerization, adding a 20-mole ratio of propylene with ethylene significantly improved the incorporation rates of dienes and reduced the incorporation rate of ethylene. The increasing order of diene incorporation rate in the E/diene/propylene polymerization was the following: VNB > ENB > VCH. In conclusion, propylene had a more stable incorporation rate compared to 1-hexene, meaning that, compared to 1-hexene, propylene actively activates the dormant ethylene sites that were inactive during the E/diene copolymerization. However, the effect of linear and cyclic dienes on *ansa*-zirconocenes/MAO catalysts has not been completely investigated, including the *ansa*-zirconocenes/TIBA/borate catalyst system.

It is well-known that the linear and cyclic dienes, by cumulative the amount of dienes in feed, as predicted, declined the catalytic activities [40]. To further investigate the effect of 1-hexene and propylene on the E/diene copolymerization, we need data on the active sites of the catalyst system. We should know how 1-hexene and propylene make a change in active centers, which is described in the later part of this paper.

In the E/ENB and E/VNB copolymerization, E/ENB had higher activity compared to E/VNB, and crystalline properties of the E/VMB copolymer are low due to the greater incorporation of VNB in the polymer. In contrast, the thermal properties were significantly higher in the presence of isoprene and butadiene, nearly equal to polyethylene, because the catalyst system failed to incorporate the isoprene and butadiene, as confirmed by ^1^HNMR (Appendix A) and DSC (Table 1). Similar effects were also noted after adding 1-hexene and propylene; however, compared to E/diene copolymerization, E/diene/1-hexane and E/diene/propylene terpolymerizations resulted in lower crystalline properties due to the higher insertion rate of 1-hexene and propylene in the system (see Table 1).

Overall, the molecular weight (Mw) of E/diene copolymers is higher than that of E/diene/1-hexene and E/diene/P except for ENB and VNB in the presence of 1-hexene (see Table 2). In contrast, isoprene and butadiene are conjugated dienes, and their effects on catalysts are stronger and lead to reduced Mw of the polymers. According to previous literature, E/diene and E/diene/P produced with the same catalyst in the presence of borate have higher Mw than produced by MMAO [37,40,41]. Among ENB, VNB, IP, VCH and BD, the ENB, VNB, and IP exhibit the electron-donating substituents in their chemical structure and present a steric hindrance that depresses the chain transfer reactions of the active centers [43]. However, the Mw of E/diene copolymer decreased with an increase of diene bulkiness, meaning that chain transfer reaction with the non-conjugated diene should be faster than with IP. In addition, the larger steric bulkiness in ENB, VNB, and VCH could be the main reason for their different chain transfer efficiencies. With lower steric hindrance of VNB and ENB with endo bond, the surrounding active site became sterically less crowded, and the rate of chain transfer became faster, resulting in E/non-conjugated diene copolymers with lower Mw. Smaller steric hindrance in BD would be favorable for fast chain transfer reaction than IP; IP with a methyl group has greater steric hindrance leading to the higher molecular weight of IP than BD. In addition, the molecular weight distributions (MWD) of E/conjugated copolymers range from 3–4, while E/non-conjugated range from 2–3; both types of copolymers are with monomodal for our catalyst, providing a clear indication of a single-site metallocene catalyst and the obtained product copolymer in nature.

Figure 3 shows the catalytic [C*]/[Zr] active centers involved in the E/diene copolymerization, E/diene/1-hexene, and E/diene/propylene terpolymerization measured at 50 °C.

The catalytic [C*]/[Zr] active centers decreased when VCH diene was used in the E/diene copolymer. In contrast, in the presence of VNB, IP, BD, and ENB, the [Zr]/[C*] ratio continuously increased and reached the maximum level. Similarly, the [Zr]/[C*] active centers ratio in E/diene/propylene polymerization is higher than E/diene polymerization, but lower than E/diene/1-hexene polymerization. During the polymerization of isoprene and butadiene, notable stability was found compared to VCH and VNB. The highest [C*]/[Zr] ratio and catalytic activity were obtained with ENB with 1-hexene. In contrast, the catalytic activity and [C*]/[Zr] ratio decreased when VNB was used, which means that VNB was considerably involved in converting the active site of PE into dormant sites. However, overall the [Zr]/[C*] ratio and catalytic activity are higher in E/diene/1-hexene and E/diene/propylene polymers compared to E/diene copolymers and show that 1-hexene and propylene activate the dormant sites that were inactive during the E/diene copolymerizations.

^1^HNMR spectroscopy is a powerful and sensitive technique used to determine the microstructure and composition of the polymer. The E/diene, E/diene/1-hexene, and E/diene/propylene polymerization microstructures are illustrated in Appendix A, respectively. In addition, Figure 4 shows the plot of E mol% in the co- and terpolymers against the active catalyst ratio.

Surprisingly, the E/VCH, EVCH/1-hexene, and EVCH/P polymerizations show a large molar ratio of E with an average active center ratio, while adding VNB to the system decreased the E molar ratio. In contrast, IP, BD, and ENB [C*]/[Zr] ratio and E mole% ratio continuously increased and reached a higher level. In addition, the mole% of ethylene is higher in the presence of 1-hexene compared to propylene, and the [C*]/[Zr] ratio is increased when propylene is used as a termonomer. In other words, propylene is more favorable than 1-hexene for activating the dormant sites of ethylene that were inactive during the E/dienes polymerization.

As shown in Figure 5, the mole% of dienes was decreased when 1-hexene was introduced in E/diene copolymers, and the [C*]/[Zr] ratio was increased and reached the maximum level of 93.44%. Similarly, VCH shows the highest mole% with 1-hexene compared to propylene. In contrast, E/ENB/1-hexene presented the maximum active center of 93.44%, with lower ENB content. The E/VNB and E/VNB/P polymers are produced with higher VNB content and a lower [C*]/[Zr] ratio, which proposes that the active catalytic sites are more functional for VNB than E and propylene.

Using the E/diene, E/diene/1-hexene, and E/diene propylene polymer composition data in Table 2, the propagation rate constants for ethylene, 1-hexene, propylene, and dienes (*k_p_*E, *k_p_*-1-H, *k_p_*PP, *k_p_*diene) were separately calculated according to our previously studied and literature [16,27,31,37,39,44].

The change in rate constant of *k_p_*E, *k_p_*-1-H, *k_p_*PP, and *k_p_*diene with k*_p_*E in the presence of different dienes under the same conditions is illustrated in Figure 6 and Table 2.

It should be noted that the *k_p_*E value of polyethylene is higher than that of the *k_p_*E of E/diene, E/diene/1-hexene, and E/diene/P polymers. To make clear comparison between the propagation rates content of E/diene copolymers and E/diene/1-hexene and E/diene/P terpolymerization, we plot the propagation data in three different plots, such as a comparison between the E mole% vs. *k_p_*E in Figure 6, [Zr]/[C*] vs. *k_p_*E in Figure 7 and, [Zr]/[C*] vs. *k_p_*diene in Figure 8. In E/diene copolymerization, the values of *k_p_*E with VNB, IP, and BD were moderate, while *k_p_*E with ENB is lower than PE. This difference was built because of the diffusion barrier in the system; as we had seen, ethylene mole% decreased when VCH and ENB were used, the copolymers were low crystalline, and a small part could be dissolved in toluene and increased the diffusion barrier, which leads to low *k_p_*E value. But the lower nearly 0 mole% of isoprene and butadiene in the copolymer and terpolymers led to improved crystalline properties and reduced the diffusion barrier, and presented a higher *k_p_*E value (see Figure 7) In contrast, E/diene/1-hexene and E/diene/P terpolymers, due to the higher incorporation rate of propylene and 1-hexene, are nearly amorphous and possibly can be dissolved in the reaction solution and increase the diffusion barrier, which leads to a lower *k_p_*E and *k_p_*diene compare to the copolymers *k_p_*E and *k_p_*diene.

Table 2 summarizes the composition, [C*]/[Zr] ratio, and propagation rate constant data, such as *k_p_*E *k_p_*1-H, *k_p_*PP, and *k_p_*diene. In Figure 7, Figure 8 and Figure 9, the *k_p_*E *k_p_*1-H, *k_p_*PP, and *k_p_*diene values and [C*]/[Zr] present very interesting comparisons for different dienes with propylene and 1-hexene. The effects of dienes, propylene, and 1-hexene are comparable with ethylene homopolymerization. In E/diene, E/diene/1-hexene, and E/diene/P, *k_p_*E is much greater then *k_p_*1-hexene, *k_p_*PP and *k_p_*diene. As shown in Figure 7, *k_p_*E of copolymers is higher than E homo polymer which means that the active center of PE, E/diene, E/diene/1-hexene, and E/diene/P are different from each other. The catalytic active centers in the copolymerization and terpolymerization produced with VCH, VNB, and BD-produced copolymers and terpolymers might be made up of loosely associated ion-pairs with large *k_p_*E, *k_p_*VNB, *k_p_*VCH, *k_p_*BD, *k_p_*1-H and *k_p_*PP values. In contrast, catalytic active centers in the co- and ter polymerization of isoprene and ENB are possible because of a large number of contact ion pairs with lower *k_p_*E, *k_p_*ENB, and *k_p_*IP values. Figure 7 and Figure 8 demonstrate the clear comparisons among the active center ratio and propagation rate constants of E/diene, E/diene/1-hexene, and E/diene/propylene polymerizations.

In Figure 9, the E/diene/1-hexene and E/diene/P, the propagation rate constant value of 1-hexene is higher than that of propylene due to the lower incorporation rate of 1-hexene, resulting polymers have more crystalline properties. In contrast, the diffusion barrier increased due to higher propylene insertion rates, resulting in lower *k_p_*PP values. In terpolymerization, the higher mole% of propylene or either 1-hexene increased the solubility of the amorphous-based polymers, leading to decreased *k_p_*E and *k_p_*diene values.

The active center Involved in E copolymerization with ENB, VNB, and VCH has a lower propagation rate than E/diene/1-hexene and E/diene/propylene terpolymers due to their higher amorphous properties. The crystalline properties of isoprene and butadiene-based polymers are increased, with low solubility leading to higher propagation rate constants. The difference in propagation rate constants between E/diene copolymers and E/diene/1-hexene and E/diene/P is mainly attributed to a diffusion barrier.

As explained in the above paragraph, the *k_p_*E of active centers developed in the E/IP, E/BD, and E/ENB copolymer process tends to be greater than those formed by E/diene/1-hexene and E/diene/propylene. Formerly, our group reported the E and P homopolymers, but their copolymers are more interesting. Similarly, it is well known that these dienes (VCH, VNB, and IP) are sterically larger than propylene and ethylene; a constituent of zirconocene precursor in the form of contact ion pairs may be unable to encourage VCH, VNB, and IP for further coordination with catalyst and leading to acting as inactive or dormant sites in the E/diene copolymerization. In addition, the [C*]/[Zr] ratio in E/diene/1-hexene and E/diene/1-hexene is higher than that of copolymers due to the lower steric hindrance of propylene and 1-hexene, suggesting that propylene and 1-hexene significantly reactivate the active sites that have been dormant during the E/diene copolymerization. Similarly, the [C*]/[Zr] ratio of previously reported ethylene homopolymerization is higher than E/VNB copolymers, which means that VNB endocyclic and exocyclic π bonds deactivate the active catalytic sites that were active during the ethylene homopolymers. In the case of butadiene, the *k*_p_*E* value of E/BD is 731 and that of E/BD/P is 451, which is lower than the copolymers; this means that the high incorporation rate of propylene increased the amorphous property and dissolved the E/BD/P terpolymer polymer in the solution leading to an increase in the diffusion barrier and resulting in a lower *k*_p_*E* value. In addition, due to the conjugation properties of butadiene, it is more problematic to incorporate in the co- or terpolymers chains and decrease the active sites with a higher propagation rate, which is nearly equal to the PE *k*_p_*E*. The dienes used in this work exhibit conjugated (isoprene and butadiene) and non-conjugated (ENB, VNB, and VCH) properties with different chemical structures and their effects on the catalyst are different. ENB, VNB, and VCH are cyclic and bulkier; ENB and VNB react with their exocyclic bonds, while VCH reacts with its exocyclic bond. In contrast, isoprene and butadiene are conjugated and less reactive for this catalyst system. This catalyst is less bulky and has low steric hindrance around the zirconium metal, allowing isoprene and butadiene to chelate easily with the Zr metal and decreasing the incorporation rate.

Similarly, 1-hexene is linear and less bulky than propylene and readily undergoes the cyclization process, decreasing the incorporation rate of dienes, and increasing the crystalline properties, leading to a higher propagation rate. Propylene exhibits a bulkier chemical structure, but due to the low steric hindrance of catalyst active sites is readily available for propylene polymerization, which produces a lower *k_p_*E. Amazingly, the E/ENB, E/ENB/1-hexene, and E/ENB/P co- and terpolymerization presents a higher active center ratio with a lower propagation rate constant compared to the E/VNB, E/VNB/1-hexene and E/VNB/P co- and terpolymerization. Even better are bicyclic olefins, but they have a different vinyl bond. The vinyl bonds of both monomers are nearly unreactive to zirconocene and Ziegler–Natta catalysts. However, VNB exhibiting a cis (cis-CH=CH–) π bond and demonstrating a more substantial ring strain appeared to be directly associated with the comonomer’s bulkiness. In addition, the VNB double may chelate with zirconium metal and prevent the insertion of ethylene, leading to a lower diffusion barrier and higher *k_p_*E.

## 6. Conclusions

In the ethylene/diene copolymers and ethylene/diene/1-hexene and ethylene/diene/propylene terpolymerizations, BD, IP, and ENB show higher activities (3-4 × 10^6^ gpoly/mol_Mt_·h), while crystalline properties are significantly higher with isoprene and butadiene (<200 J/g), nearly equal to polyethylene. In contrast, E/diene/1-hexane and E/diene/propylene terpolymerizations have lower crystallinity (>10 J/g) due to the higher insertion rate of 1-hexene and propylene. The [C*]/[Zr] ratio decreased when VCH was added, while VNB, IP, BD, and ENB caused a significant increase in the order VNB < IP < BD < ENB. After adding a 20-mole ratio of propylene, the [Zr]/[C*] ratio increased, reaching a higher level of 97%, which is higher than E/diene polymerizations, but it was significantly reduced when 1-hexene was used as a termonomer, indicating that the active center [C*]/[Zr] ratio strongly depended on the type of monomers. In E/diene copolymerization and E/diene/1-hexene and E/diene/propylene, the propagation rate constant (*k_p_*E) with VNB, IP, and BD was moderate, while the *k_p_*E with ENB was lower than PE. This difference arises because of the diffusion barrier in the system. Although E mole% decreased when VCH and ENB were used, the copolymers exhibited low crystallinity, and a small part of polymer could be dissolved in toluene and increased the diffusion barrier, which leads to a low *k_p_*E value. The *k_p_* values of dienes, especially VNB in the E/VNB copolymers and E/VNB/1-hexene and E/VNB/propylene terpolymers, are similar, but the insertion rate of VNB is much lower in E/VNB/P terpolymers, meaning that active catalytic centers in the copolymers are different from those in the terpolymers. In contrast, the lower (nearly 0 mole%) of isoprene and butadiene in the copolymer and terpolymers led to improved crystalline properties, reduced diffusion barriers, and presented a higher *k_p_*E value.

## Figures and Tables

**Figure 1 polymers-14-03239-f001:**
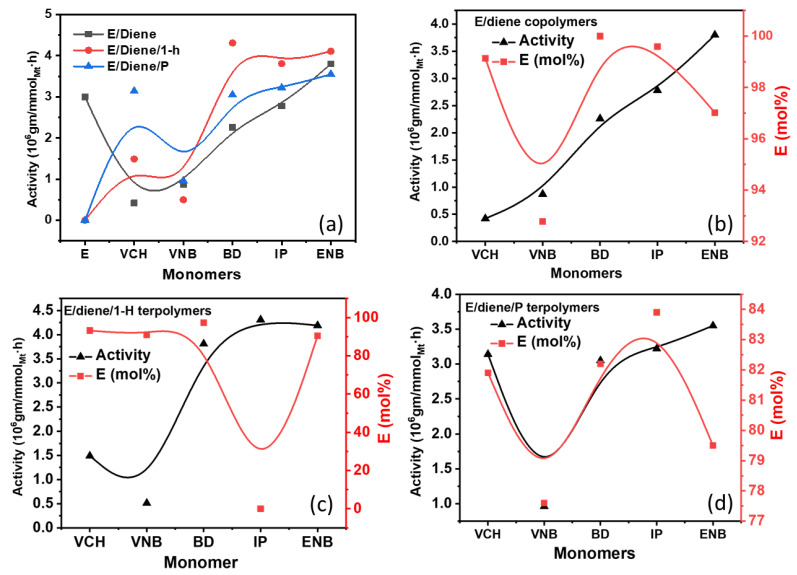
(**a**) Graph of polymerization activity between E/diene, E/diene/1-hexene and E/diene/propylene polymerizations, (**b**) comparison of E content and activity in E/diene copolymers, (**c**) comparison of E content and activity in E/diene/1-hexene terpolymers, (**d**) comparison of E content and activity in E/diene/propylene terpolymers.

**Figure 2 polymers-14-03239-f002:**
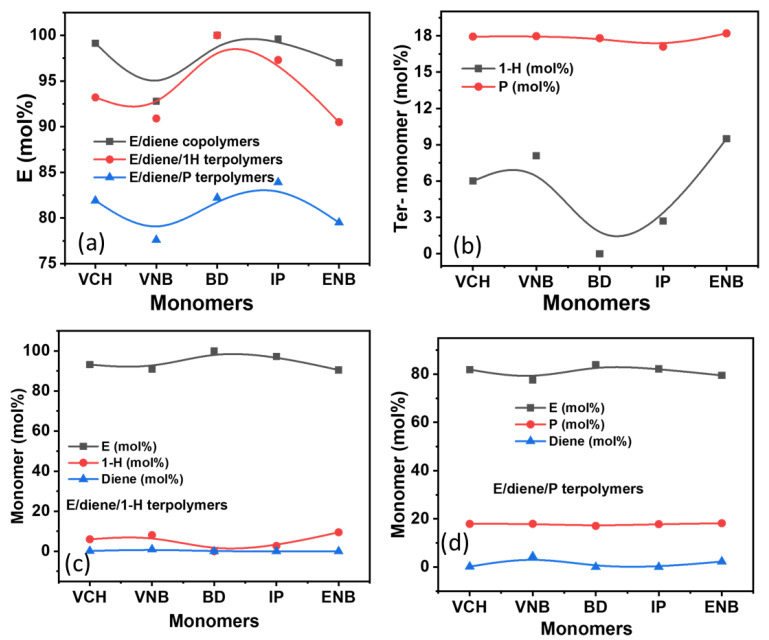
(**a**) Plots of E content in E/diene, E/diene/1-hexene, and E/diene/propylene polymerizations, (**b**) content of propylene and 1-hexene in E/diene/1-hexene and E/diene/propylene, (**c**) comparison of the content of E, diene, and 1-hexene, (**d**) comparison of the content of E, diene, and propylene.

**Figure 3 polymers-14-03239-f003:**
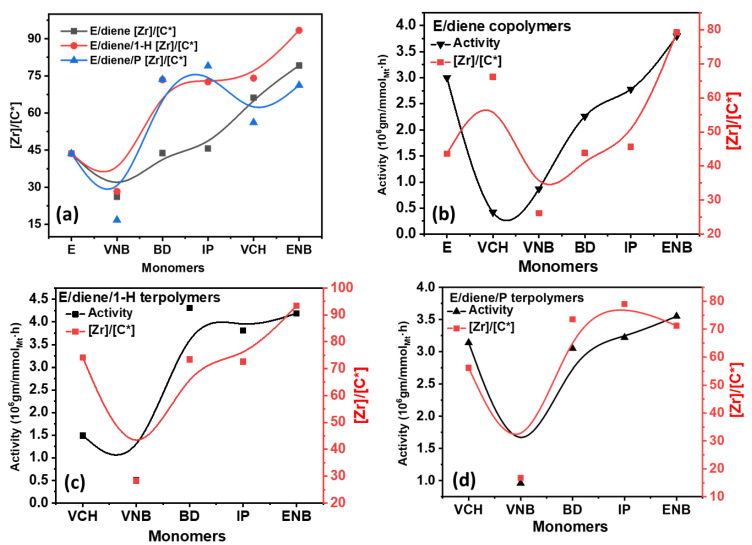
(**a**) Comparison of [Zr]/[C*] ratio between E/diene, E/diene/1-hexene and E/diene/P polymerizations, (**b**–**d**) are correlations between [Zr]/[C*] ratio and catalytic activities in E/diene, E/diene/1-hexene, and E/diene/propylene polymerizations, respectively.

**Figure 4 polymers-14-03239-f004:**
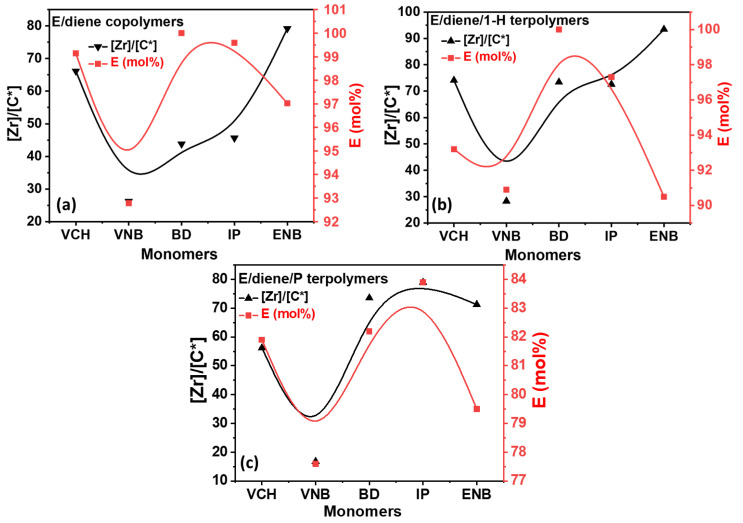
(**a**) Comparison between active centers and ethylene mole% in E/diene copolymers; (**b**,**c**) comparison of active centers and ethylene mole% in E/diene/1-hexene and E/diene/P terpolymerizations.

**Figure 5 polymers-14-03239-f005:**
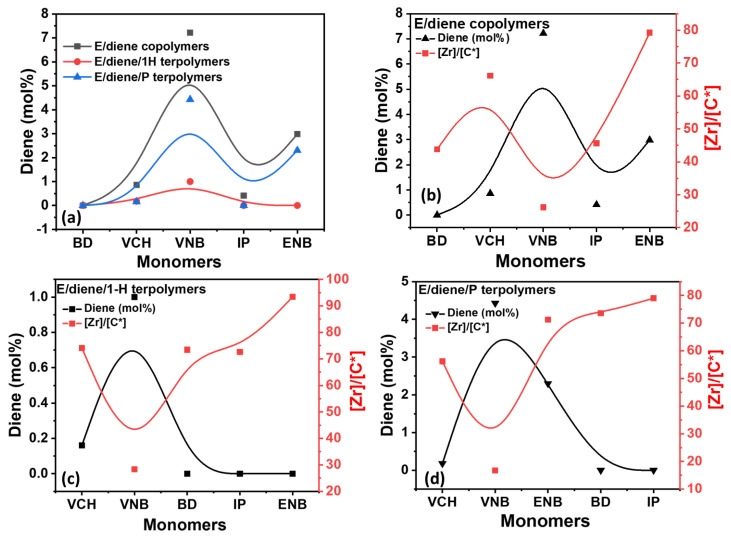
(**a**) Comparison between active centers and diene mole% in E/diene copolymers while (**b**–**d**) is the comparison of active centers and diene mole% in E/diene copolymers and E/diene/1-hexene and E/diene/P terpolymerizations.

**Figure 6 polymers-14-03239-f006:**
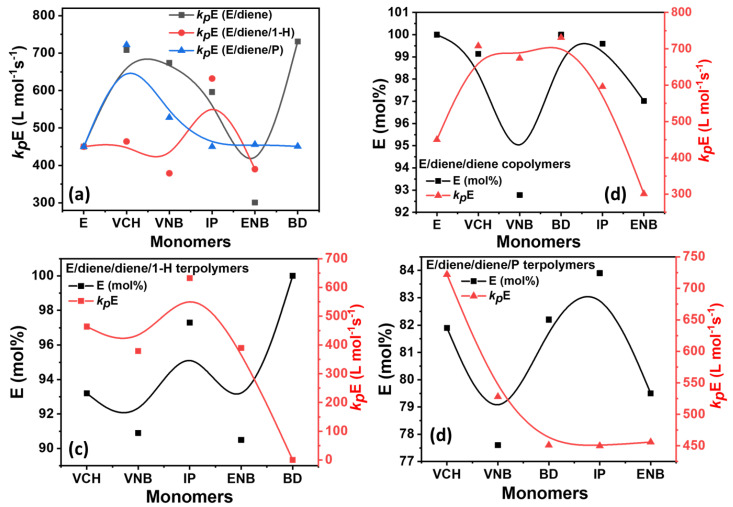
(**a**) Comparison of *k_p_*E value of E/diene, E/diene/1-hexene and E/diene/P polymerizations; (**b**–**d**) correlations between *k_p_*E value and E mol% in E/diene, E/diene/1-hexene, and E/diene/propylene polymerizations, respectively.

**Figure 7 polymers-14-03239-f007:**
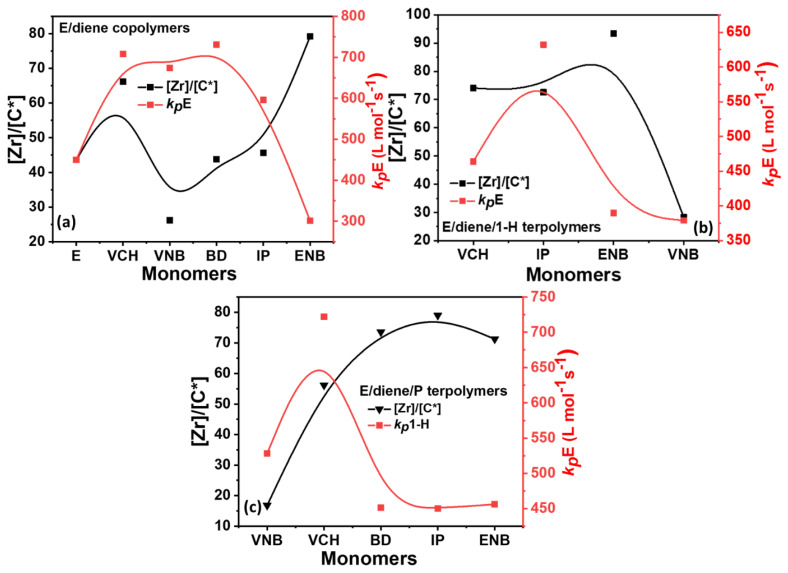
(**a**) relationships between active center and *k_p_*E in E/diene, while (**b**,**c**) are relationships between active center and *k_p_*E in E/diene/1-hexene and E/diene/propylene polymerizations, respectively.

**Figure 8 polymers-14-03239-f008:**
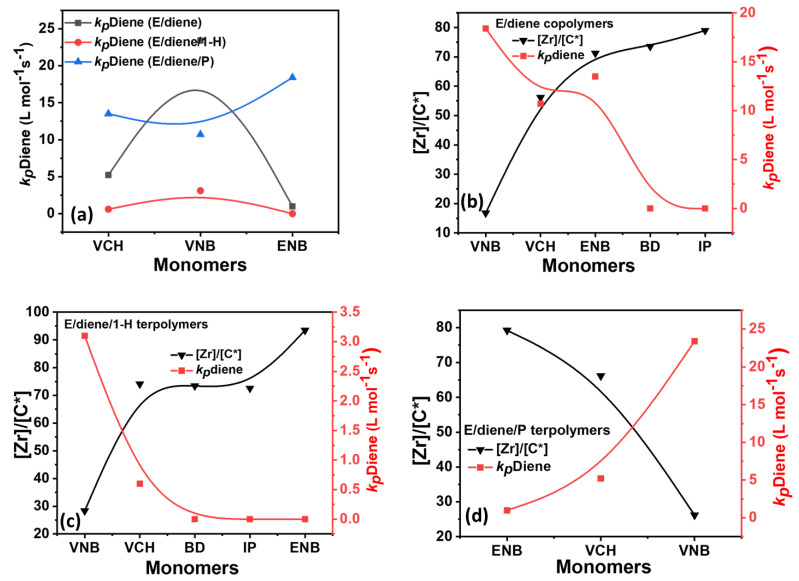
(**a**) Comparison of *k_p_*diene value of E/diene, E/diene/1-hexene and E/diene/P polymerizations while (**b**–**d**) relationships between active center and *k_p_*diene in E/diene, E/diene/1-hexene and E/diene/propylene polymerizations, respectively.

**Figure 9 polymers-14-03239-f009:**
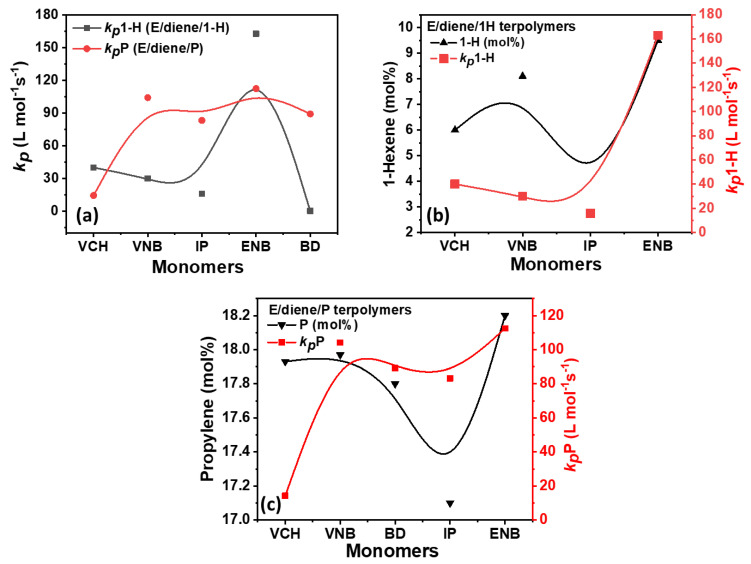
(**a**) Comparison of *k_p_*1-H and *k_p_*P in E/diene/1-hexene and E/diene/P polymerizations, (**b**) relationship between 1-hexene mol% and *k_p_*1-H in E/diene/1-hexene terpolymerization, (**c**) relationship between propylene mol% and *k_p_*P in E/diene/propylene polymerizations.

**Table 1 polymers-14-03239-t001:** E/diene, E/diene/1-hexene and E/diene/propylene polymerization catalyzed with (*rec*-Et(Ind)_2_ZrCl_2_)/(Ph_3_C)B(C_6_F_5_)_4_/TIBA catalyst system ^a^.

Run	Dienes	Activity ^b^	Ethylene ^c^	Diene ^c^	1-Hexene ^c^	Propylene ^c^	[Zr]/[C*] ^d^	*T*m ^e^	Δ*H*m ^e^
		×10^6^	mol%	mol%	mol%	mol%		(°C)	(J/g)
1.1	E	3.00	100				43.6	132	202
1.2	VCH	0.42	99.14	0.86			66.17	131.7	112.7
1.3	VNB	0.87	92.78	7.22			26.17	118.8	8.9
1.4	BD	2.26	100	0.00			43.79	131.7	214.9
1.5	IP	2.78	99.59	0.41			45.61	128.2	168.0
1.6	ENB	3.80	97.02	2.98			79.2	126.6	74.7
1.7	VCH	1.49	93.2	0.16	6.0		74.1	92.6	58.8
1.8	VNB	0.51	90.9	1.00	8.1		28.3	83.6	17.5
1.9	BD	4.31	100	00	Cyclization		73.42	118.1	29.9
1.10	IP	3.81	97.3	00	2.7		72.58	111.4	34.1
1.11	ENB	3.80	97.02	0.00	9.5		93.44	64.6	6.1
1.12	VCH	3.14	81.9	0.18		17.93	56.16	92.5	58.8
1.13	VNB	0.96	77.6	4.43		17.97	16.76	105.6	25.5
1.14	BD	3.05	82.2	N/A		17.8	73.56	101.3	7.2
1.15	IP	3.22	83.9	N/A		17.1	78.99	103.1	3.8
1.16	ENB	3.55	79.5	2.30		18.2	71.23	108.4	22.0

^a^ Reaction conditions: Catalyst 1.25 mole, TIBA 1000 μmole, borate 2.5 μmole, and quenching agent TPCC 2000 μmole, ^b^ (10^6^ gm/mmol_Mt_·h). ^c^ Calculated by ^1^HNMR; ^d^ Active center [Zr]/[C*] determined through the sulfur analyzer; ^e^ Determined by DSC.

**Table 2 polymers-14-03239-t002:** E/diene, E/diene/1-hexene, and E/diene/propylene polymerization catalyzed with (*rec*-Et(Ind)_2_ZrCl_2_)/(Ph_3_C)B(C_6_F_5_)_4_/TIBA catalyst system ^a^.

Run	Dienes	[Zr]/[C*] ^b^	*k* _p_ *E* ^c^	*k* _p_ *dienes* ^d^	*k* _p_ *1-H* ^e^	*k* _p_ *P* ^f^	Mw ^g^	Ɖ
		%	L mol^−1^·s^−1^	L mol^−1^·s^−1^	L mol^−1^·s^−1^	L mol^−1^·s^−1^	Kg/mole	
1.1	E	43.6	450				88	44
1.2	VCH	66.17	708	5.228			82	4.77
1.3	VNB	26.17	674	23.395			25	3.51
1.4	BD	43.79	731				80	4.12
1.5	IP	45.61	596				65	3.52
1.6	ENB	79.2	301	0.987			89	3.82
1.7	VCH	74.1	464		40		56.30	2.66
1.8	VNB	28.3	379		30		52311	2.20
1.9	BD	73.42	468					
1.10	IP	72.58	632		15.8			
1.11	ENB	93.44	390		162.9		42414	1.93
1.12	VCH	56.16	722	10.7		14.3	51.0	2.79
1.13	VNB	16.76	528	18.4		104.2	23.5	3.50
1.14	BD	73.56	451			89.2		
1.15	IP	78.99	450			83.3	39.8	2.38
1.16	ENB	71.23	456	13.5		112.5	35.3	2.52

**^a^** Reaction conditions: Catalyst 1.25 mole, TIBA 1000 μmole, borate 2.5 μmole, and quenching agent TPCC 2000 μmole. ^b^ Active center [Zr]/[C*] determined through the sulfur analyzer; ^c^ propagation constant of ethylene; ^d^ propagation constant of dienes; ^e^ propagation constant of 1-hexene; ^f^ propagation constant of propylene; ^g^ Determined by GPC.

## Data Availability

Not applicable.

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
