# Peer review of "Progress toward Polymerization Reaction Monitoring with Different Dienes: How Small Amounts of Dienes Affect ansa-Zirconocenes/Borate/Triisobutylaluminium Catalyst Systems"

_polymers, 2022, doi:10.3390/polym14163239_

Round 1
Reviewer 1 Report
This paper is an interesting study that is likely to be well cited. However, a minor revision is needed for publication. It cannot be published in the present form.
The introduction section should contain a specific hypothesis of the study (2-3 sentences)
The conclusion should contain a final remark whether the authors confirmed their hypothesis
Sometimes the font size changes suddenly. For example, line 32 and many others. This must be corrected.
The polymerization paragraph should describe the experimental procedure in detail, not by simple references to your own previous works. Why in such a small section “Polymerization” are there 4 references to your own previous work? The answer is simple: to improve your own citation metrics! Remember! The reader does not need your self-citation! The reader wants to see a qualitatively described experimental procedure, described in the smallest details. You have ignored the interests of the reader! Instead of a detailed reproducible procedure, you have provided 4 references to your own articles! I don't mind your self-citing. I object to the fact that you do not take into account the interests of the reader. My main question to the authors - did you write this article for the reader or for yourself? The second question for the authors is - how do you evaluate such a description of experimental procedures? Is it convenient for the reader? The third question is - how do you, specifically you, look at the fact that there are 13 members in your team of authors?
Thus, I repeat that this is a good study. But it must be properly formatted for publication. Therefore, it needs a minor revision.
Author Response
Reviewers -1
Comments and Suggestions for Authors
This paper is an interesting study that is likely to be well cited. However, a minor revision is needed for publication. It cannot be published in the present form.
The introduction section should contain a specific hypothesis of the study (2-3 sentences)
The conclusion should contain a final remark whether the authors confirmed their hypothesis
Sometimes the font size changes suddenly. For example, line 32 and many others. This must be corrected.
The polymerization paragraph should describe the experimental procedure in detail, not by simple references to your own previous works. Why in such a small section “Polymerization” are there 4 references to your own previous work? The answer is simple: to improve your own citation metrics! Remember! The reader does not need your self-citation! The reader wants to see a qualitatively described experimental procedure, described in the smallest details. You have ignored the interests of the reader! Instead of a detailed reproducible procedure, you have provided 4 references to your own articles! I don't mind your self-citing. I object to the fact that you do not take into account the interests of the reader. My main question to the authors - did you write this article for the reader or for yourself? The second question for the authors is - how do you evaluate such a description of experimental procedures? Is it convenient for the reader? The third question is - how do you, specifically you, look at the fact that there are 13 members in your team of authors?
Thus, I repeat that this is a good study. But it must be properly formatted for publication. Therefore, it needs a minor revision.
Response: Thank you very much for the deep insight and recommendation towards the acceptance of the manuscript after minor revision. We have carefully revised the whole manuscript as per suggestions and recommendations.
- The polymerization paragraph should describe the experimental procedure in detail, not by simple references to your own previous works. Why in such a small section “Polymerization” are there 4 references to your own previous work? The answer is simple: to improve your own citation metrics! Remember! The reader does not need your self-citation! The reader wants to see a qualitatively described experimental procedure, described in the smallest details. You have ignored the interests of the reader! Instead of a detailed reproducible procedure, you have provided 4 references to your own articles! I don't mind your self-citing. I object to the fact that you do not take into account the interests of the reader. My main question to the authors - did you write this article for the reader or for yourself? The second question for the authors is - how do you evaluate such a description of experimental procedures? Is it convenient for the reader?
Response All copolymerization and terpolymerization were performed in the 100-Ml Schlenk round bottom glass reactor under a nitrogen atmosphere. Before starting the polymerizations, the glass reactor was perched with nitrogen and added 50ml of toluene (polymerization grade). Metallocene complex and borate were prepared in 20ml of toluene under a nitrogen environment and used for the set of experiments. For E/diene copolymerization, a pure form of E was added to the reactor, followed by additions of dienes and TIBA and run for 5 min, and then metallocene and borate activator to initiate polymerizations. However, for E/diene/1-hexene polymerization, ethylene with 0.12 mol/L of 1-hexene was added. In addition, the E/diene/propylene terpolymerization, E/P 80/20 mole ratio was added and followed the same method to add the diene, TIBA, metallocene, and borate to start the polymerization. After a planned time, the E/diene copolymers and E/diene/1-hexene and E/diene/propylene polymerization reactions were quenched with TPPC (TIBA/TPCC (1000/2000 µ mole) which was selectively quenching the metal-polymer bonds through acyl chloride. In addition, further volume of ethanol with 2 % hydrochloric acid was added to decompose the borate and metallocene catalyst. The obtained polymer product was further precipitated by adding 200ml of ethanol. To completely remove the unreactive TPPC and other impurities from polymers, samples were thoroughly purified by a three-step process by dissolving the 30mg of polymer in 50 ml of octane at 120 ℃ and then precipitated in 200 ml of ethanol. [11, 31, 39] All polymer samples were dried using the vacuum dried system at 40 ℃.
Q.The second question for the authors is - how do you evaluate such a description of experimental procedures?
Response: To determine the effect of propylene and 1-hexene addition, the E/diene copolymers and E/diene/1-hexene and E/diene/propylene polymerization reactions were quenched with TPPC (TIBA/TPCC (1000/2000 µ mole) which was selectively quenching the metal-polymer bonds through acyl chloride, which has been verified in Guo et al studied in the polymerization of ethylene and propylene homo polymerization and their copolymerization catalyzed with metallocene/MMAO catalyst system. In addition, Zhang et al., has been verified the TPCC in the polymerization of olefins with heterogeneous Ziegler-Natta catalyst.
The third question is - how do you, specifically you, look at the fact that there are 13 members in your team of authors?
Response:
We have discussed the progress toward polymerization reaction monitoring with different dienes: in what way small amount of dienes affect ansa-zirconocenes/borate/triisobutylaluminium catalyst system. Firstly, A set of E/dienes copolymerizations, and ethylene/dienes/1-hexene and ethylene/dienes/1-hexene terpolymerizations catalyzed with zirconocenes/borate/triisobutylaluminium(racEt(Ind)2ZrCl2/-[Ph3C][B(C6F5)4]/triisobutylaluminium(TIBA) were performed in toluene at 50C, and each polymer propagation chain ends quenched with 2-thiophenecarbonyl. Secondly, the active center variation in the copolymerization of E with different dienes and their terpolymerization with 1-hexene and propylene were determined by using sulfur analyzer. Thirdly, the thermal properties, composition, molecular weight, and polydispersity were also studied in this work. To elevated the catalyst behaviour and obtained polymer (E/diene copolymers and E/diene/1-hexene and E/diene/propylene) properties we required different characterization such as high-temperature GPC, high-temperature NMR, ultraviolet fluorescence sulfur analyzer, DSC and quenched labeling of propagation chain experimental protocol. To complete this task in a precise way we work as a team. However, Amjad Ali, Jamile Mohammadi Moradian, Ahmad Naveed, Fan Zheqing, Li Guo conceived of the presented idea. Tariq Aziz, Muhammad Nadeem, Chanez Maouche, and Yintian Guo developed the theory and performed the computations. Ahmad Naveed, Fan Zheqing and Li Guo verified the analytical methods. Fan Zheqing and Li Guo encouraged Amjad Ali, Jamile Mohammadi Moradian, Ahmad Naveed, Tariq Aziz, Muhammad Nadeem, Chanez Maouche, Yintian Guo, Waleed Yaseen , Maria Yassen, Fazal Haq, Mobashar Hassan to investigate the progress toward polymerization reaction monitoring with different dienes: in what way small amount of dienes affect ansa-zirconocenes/borate/triisobutylaluminium catalyst system. ” and supervised the findings of this work. All authors discussed the results and contributed to the final manuscript. Amjad Ali, Jamile Mohammadi Moradian, Fan Zhiqiang and Li Guo took the lead in writing the manuscript all authors provided critical feedback and helped shape the research, analysis and manuscript.

Reviewer 2 Report
The manuscript under consideration by Ali et al entitled “Progress toward polymerization reaction monitoring with different dienes: in what way small amount of dienes effect ansazirconocenes/botate/triisobutylaluminium catalyst system” is thoroughly and critically studied. The manuscript is well-organized and written. The manuscript experimentally shows the effect of several different parameters on the efficiency of ansazirconocenes/botate/triisobutylaluminium catalyst system for diene structure and polymer chain propagation. Several dienes were used as comonomers such as VCH, VNB, IP, BD, ENB etc. In principle, I recommend the publication of this manuscript after few important points to consider
1. At some points expalination is not enough and author should elaborate more in text. As a typical example, captions of Figures are not very informative along with insufficient explanation in the text, see Figure 1 just an example. Others are same too.
2. Abstract and conclusion should be made sufficiently different. Abstract mainly should contain what has been done while conclusion should contain the important outcomes of the study.
3. Only data is presented in Tables and plots. I would strongly recommend to show all the chromatograms and spectra that are used to take data to conclude, as supplementary material.
Author Response
Reviewers -2
Comments and Suggestions for Authors
The manuscript under consideration by Ali et al entitled “Progress toward polymerization reaction monitoring with different dienes: in what way small amount of dienes effect ansazirconocenes/botate/triisobutylaluminium catalyst system” is thoroughly and critically studied. The manuscript is well-organized and written. The manuscript experimentally shows the effect of several different parameters on the efficiency of ansazirconocenes/botate/triisobutylaluminium catalyst system for diene structure and polymer chain propagation. Several dienes were used as comonomers such as VCH, VNB, IP, BD, ENB etc. In principle, I recommend the publication of this manuscript after few important points to consider
Response: Thank you very much for the deep insight into improving the quality of the manuscript and recommendation towards the acceptance of the manuscript after minor revision. We have carefully revised the whole manuscript as per suggestions and recommendations.
- At some points expalination is not enough and author should elaborate more in text. As a typical example, captions of Figures are not very informative along with insufficient explanation in the text, see Figure 1 just an example. Others are same too.
Response: Thank you very much for your suggestion. The required changes have been made in the revised manuscript text, and also revised all figures captions according to your suggestion, for example,
Figure 1, (a) the graph of polymerization activity between E/diene, E/diene/1-hexene and E/diene/propylene polymerizations, (b) comparison of E content and activity in E/diene copolymers (c) comparison of E content and activity in E/diene/1-hexene terpolymers (d) comparison of E content and activity in E/diene/propylene terpolymers.
Figure 4 (a) Comparison between active centers and ethylene mole % in E/diene while (b) and (c) is the comparison of active centers and ethylene mole % in E/diene/1-hexene and E/diene/P terpolymerizations
Figure 5 (a) Comparison between active centers and diene mole % in E/diene copolymers, while (b) and (c) is the comparison of active centers and diene mole % in E/diene/1-hexene and E/diene/P terpolymerizations.
Figure 6 (a) compression of kpE value of E/diene, E/diene/1-hexene and E/diene/P polymerizations while (b), (c) and (d) are correlations between kpE value and E mol% in E/diene, E/diene/1-hexene and E/diene/propylene polymerizations, respectively.
Figure 7 (a) relationships between active center and kpE in E/diene, while (b) and (c) are relationships between active center and kpE in E/diene/1-hexene and E/diene/propylene polymerizations, respectively.
Figure 8 (a) Compression of kpdiene value of E/diene, E/diene/1-hexene and E/diene/P polymerizations while (b), (c) and (d) relationships between active center and kpdiene in E/diene, E/diene/1-hexene and E/diene/propylene polymerizations, respectively.
Figure 9 (a) Compression of kp1-H and kpP in E/diene/1-hexene and E/diene/P polymerizations (b) relationship of 1-hexene mol% and kp1-H in E/diene/1-hexene terpolymerization (c) relationship of propylene mol% and kpP in E/diene/propylene polymerizations.
- Abstract and conclusion should be made sufficiently different. Abstract mainly should contain what has been done, while the conclusion should contain the important outcomes of the study.
Response: Thank you very much for your suggestion.
Abstract
The objectives of this work were to address the fundamental characteristics of ansa-zirconocene catalyzed E/diene copolymerization and E/diene/1-hexene and E/diene/propylene terpolymerizations, the quantitative relationship between dienes structure and polymer chain propagation rate constant in term of quantifiable catalytic active sites. One of the most important but unknown factors in olefins ansa-zirconocene complexes is the distribution of the catalyst between the actively participating in polymer chain formation and dormant sites. A set of ethylene/dienes copolymerizations, and ethylene/dienes/1-hexene and ethylene/dienes/1-hexene terpolymerizations catalyzed with ansa-zirconocenes/borate/triisobutylaluminium (rac-Et(Ind)2ZrCl2/[Ph3C][B(C6F5)4]/triisobutylalumin-ium(TIBA) were performed in toluene at 50C. To determine the active center [C*]/[Zr] ratio variation in the copolymerization of E with different dienes and their terpolymerization with 1-hexene and propylene, each polymer propagation chain ends quenched with 2-thiophenecarbonyl, which selectively quenching the metal-polymer bonds through acyl chloride. The ethylene, propylene, 1-hexene, and dienes composition-based propagation rate constants (kpE, kpP, kp1-H and kpdiene) thermal (melting and crystalline) properties, composition (mol% of ethylene, propylene, 1-hexene and diene), molecular weight, and polydispersity were also studied in this work. Systematic comparisons of the proportion of catalytically [Zr]/[C*] active sites and polymerization rate constant (kp) for ansa-zirconocenes catalyzed E/diene, E/diene/1-hexene, and E/diene/propylene polymerization have not been reported before. We evaluated the addition of 1-hexene and propylene as termonomer in the copolymerization E/diene. To make a comparison for each diene under identical conditions, we started the polymerization by introducing an 80/20 mole ratio of E/P and 0.12 mol/L of 1-hexene in the system. The catalyst behavior against different dienes, 1-hexene, and propylene, is very interesting such as a change in thermal properties, cyclization of 1-hexene, and decreased incorporation of isoprene and butadiene; how to change the defusion barriers in the system and its effect on kp.
Conclusion
In the Ethylene/diene copolymers and ethylene/diene/1-hexene and ethylene/diene/propylene terpolymerizations, BD, IP, and ENB show higher activities (3-4x106gpoly/molMt*h), while crystalline properties are significantly higher with isoprene and butadiene (<200 J/g), nearly equal to polyethylene. In contrast, E/diene/1-hexane and E/diene/propylene terpolymerizations showed lower crystalline (>10 J/g), properties due to the higher insertion rate of 1-hexene and propylene. The [C*]/[Zr] ratio decreased when VCH was added, while VNB, IP, BD, and ENB significantly increased in the following order VNB < IP < BD < ENB. After adding the 20-mole ratio of propylene, the [Zr]/[C*] ratio increased and reached a higher level of 97%, which is higher than E/diene polymerizations, but it was significantly reduced when 1-hexene was used as a termonomer, which mean that active center [C*]/[Zr] ratio strongly depended on the type of monomers. In E/diene copolymerization and E/diene/1-hexene and E/diene/propylene, the propagation rate constant (kpE) with VNB, IP, and BD was moderate, while kpE with ENB was lower than PE. This difference is built because of the diffusion barrier in the system. While E mole % decreased when VCH and ENB were used, the copolymers exhibited low crystalline, and a small part of polymer could be dissolved in toluene and increased the defusion barrier, which leads to a low kpE value. The kp values of dienes, specially VNB in the E/VNB copolymers and E/VNB/1-hexene and E/VNB/propylene terpolymers, are nearly close, but the insertion rate of VNB is much lower in E/VNB/P terpolymers, meaning that active catalytic centers in the copolymers are different from the terpolymers. In contrast, the lower nearly 0 mole % of isoprene and butadiene in the copolymer and terpolymers led to improved crystalline properties, reduced defusion barriers, and presented a higher kpE value.
- Only data is presented in Tables and plots. I would strongly recommend showing all the chromatograms and spectra that are used to take data to conclude as supplementary material.
Thank you very much for your valuable suggestions for improving our manuscript. We illustrated all polymers spectra in the supporting formation Figures S1a- S1e, S2a- S2e and S3a- S3e. These were used to determine each monomer's composition and propagation rate constant.
